# Identification and comprehensive analysis of super-enhancer related genes involved in epithelial-to-mesenchymal transition in lung adenocarcinoma

**Yifei Liu** [iD] *

Clinical Center for Molecular Diagnosis and Therapy, The Second Affiliated Hospital of Fujian Medical University, Quanzhou, Fujian Province, China

* 2217579236@qq.com

## Abstract

Lung adenocarcinoma is a disease with a high mortality rate, and its mechanism is still unclear. Super-enhancers play an important role in gene expression and also affect the occurrence and development of lung adenocarcinoma, so more and more people pay attention to them. In order to explore the influence of super-enhancer related genes on tumor development, we identified super-enhancer regulated genes related to Epithelial-to-mesenchymal transition (EMT). By analyzing the single-cell sequencing data and the TCGA database of lung adenocarcinoma, we suggest that the up-regulation of *TMSB10* in lung adenocarcinoma and its association with poor prognosis may be due to the regulation of super-enhancers during tumor cell metastasis. Using the TCGA lung adenocarcinoma data set, the samples were divided into *TMSB10* high-expression group and low-expression group, and it was found that there were significant differences in immune infiltration between the high-expression group and the low-expression group. We parted 513 samples into eight *TMSB10*-related molecular subtypes using differentially expressed genes of high and low *TMSB10* expression groups. We concentrated on four molecular subtypes with the most significant clusters, each with its own characteristics in terms of Immune cell infiltration, prognosis, or pathological stages. In order to predict the four molecular subtypes, we established a prediction model using random forest, and the external test results showed that the prediction accuracy of the model was 0.87. This study may provide potential help for the study of the mechanism of metastasis and invasion of lung adenocarcinoma cells and personalized treatment of lung adenocarcinoma.

## Introduction

Lung adenocarcinoma is a disease with a very high mortality rate and lung adenocarcinoma is the most common pathological type of lung cancer [1, 2]. So far, the mechanism of its occurrence and development remains unclear. Epithelial-to-mesenchymal transition (EMT) is involved in a variety of lung diseases, such as idiopathic pulmonary fibrosis and lung

**Data Availability Statement:** TCGA:https://portal.gdc.cancer.gov/repository?facetTab=cases&filters=%7B%22op%22%3A%22and%22%2C%22content%22%3A%5B%7B%22op%

22%3A%22in%22%2C%22content%22%3A%7B
%22field%22%3A%22cases.project.program.
name%22%2C%22value%22%3A%5B%22TCGA
%22%5D%7D%7D%2C%7B%22op%22%3A%
22in%22%2C%22content%22%3A%7B%22field
%22%3A%22cases.project.project_id%22%2C%
22value%22%3A%5B%22TCGA-LUAD%22%5D
%7D%7D%7D%5D%7D GEO:https://www.ncbi.nlm.
nih.gov/geo/query/acc.cgi?acc=GSE131907.

**Funding:** This work was funded by grants from Natural Science Foundation of Fujian Province (Grant number: 2019J01474). The funders had no role in study design, data collection and analysis, decision to publish, or preparation of the manuscript.

**Competing interests:** The authors have declared that no competing interests exist.

adenocarcinoma. Studies have shown that the metastasis and invasion of cancer cells are closely connected with EMT [3, 4]. Inhibiting the EMT process of cancer cells will help the prognosis of lung adenocarcinoma patients. EMT are divided into three types: type-1 is associated with embryonic development, type-2 with organ fibrosis and tissue regeneration, and type-3 with tumor progression and metastasis [5].

Super-enhancers(SE) are key regulatory elements that control tissue-specific transcription [6]. Super-enhancers were identified by locating H3K27ac signal in ChIP-seq. Compared with typical enhancers, super-enhancers have a higher density of transcription factors, and genes regulated by super-enhancers are often highly expressed [7]. Super-enhancers have been identified as the basic oncogenes driver needed to maintain the identity of cancer cells. Studies have shown that cancer cells produce new super-enhancers related genes(SE-genes) that are involved in the pathogenesis of tumors, and if these SEs are missing, the survival rate of cancer cells is significantly reduced [8]. In addition, pharmacological inhibitors targeting fundamental components of super enhancer assembly and activation have shown great promise for reducing tumor growth and proliferation in several preclinical tumor models [9, 10].

Although super-enhancers associated with lung adenocarcinoma have been extensively studied, super-enhancers affecting EMT associated with metastasis and invasion of lung adenocarcinoma cells have received little attention. Therefore, we focused on EMT-related super-enhancers and SE-genes to investigate their role in lung adenocarcinoma.

Combined with single-cell RNA-seq and TCGA database analysis, we identified and focused on the SE-gene *TMSB10* involved in EMT in lung adenocarcinoma. We investigated the biological significance of *TMSB10* expression levels, identified and characterized the molecular subtypes associated with *TMSB10*.

## Materials and methods

### Flowchart of the workflow

In order to more intuitively introduce our research, we summarize the workflow in Fig 1.

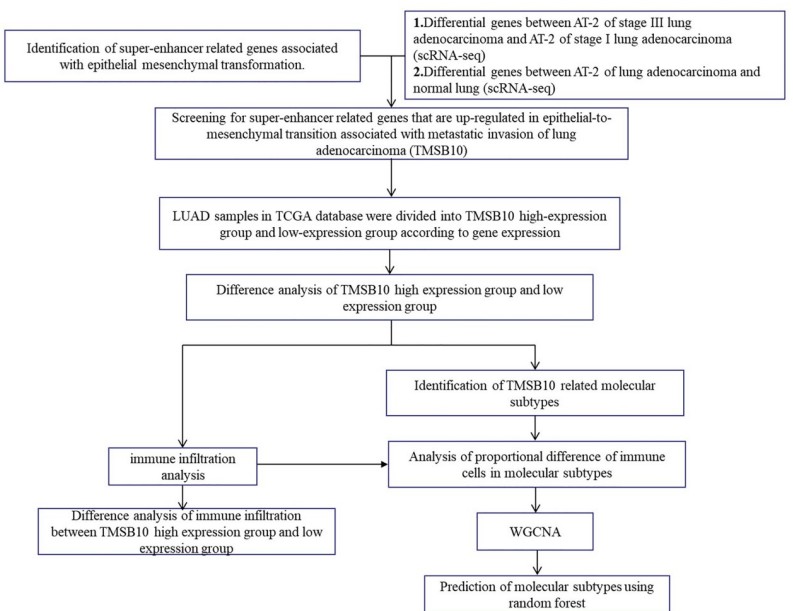

**Fig 1. Flowchart of the workflow.**

## Data source

The data for this study were derived from lung adenocarcinoma expression data from the TCGA and single cell RNA-seq data(GSE131907) from GEO database [11]. Super-enhancers identification results of A549 before and after TGF-β induction were obtained from SEdb 2.0 [12]. We analyzed the results of super-enhancer identification before and after TGFβ-induced A549 in SEdb 2.0 and studied the active genes closest to super-enhancers as super-enhancer related genes(SE-genes). The information on the data sets we used lists in Table 1.

## Single cell RNA-seq analysis

We used R-package Seurat to analyze single-cell RNA-seq data [13], NormalizeData() to normalize the data, FindVariableFeatures() to find variable features. We use FindIntegrationAnchors() to find anchors, and IntegrateData() to integrate samples and correct batch effects. We utilize FindMarkers () to analyze differential genes. We used cell markers to identify lung epithelial cell subtypes and focused on the characterization of alveolar epithelial cell typeII(AT-2) in stage I and Stage III lung adenocarcinoma. In order to observe the most features, the threshold of difference analysis was relaxed and set at p value<0.05. Cell markers for the epithelial cell subtypes are given in Table 2. We used R package copykat to identify cancer cells.

## Difference analysis

We used R package DEseq2 [14] to calculate the difference of *TMSB10* expression between high and low groups in lung adenocarcinoma data of TCGA. The high and low groups were characterized by the medians of the expression after the variance-stabilizing transformation (VST). The expression levels of EMT-related SE-genes and survival analysis results of lung adenocarcinoma were obtained from UALCAN [15].

## GO and KEGG pathway enrichment analysis

We use DAVID (https://david.ncifcrf.gov/home.jsp, version 6.8) [16, 17] for analysis of KEGG pathways and GO, choose P value<0.01 items to display.

## Research on immune cell infiltration

The TCGA lung adenocarcinoma count expression matrix normalized by R packet DEseq2 (VST) was calculated by CIBORSORT for immune cell infiltration. The difference in cell proportion between the high and low TMSB10 expression groups was calculated using wilcoxon rank sum test.

## Identification and characterization of *TMSB10* related molecular subtypes in lung adenocarcinoma

We used R package DEseq2 [14] to normalize the expression matrix of differential genes in the high-expression and low-expression groups of *TMSB10*, P value<0.05 was used as the

**Table 1. Information on data sets.**

| Sample number | platform | type | sample count |
|---|---|---|---|
| Case:ENCSR783SNV Input:ENCSR838ZAU | Illumina HiSeq 2000 | ChIP-seq | 1 |
| Case: GSM2406904 Input: GSM2406904 | Illumina HiSeq 2000 | ChIP-seq | 1 |
| GSE131907 | Illumina HiSeq 2500 | ScRNA-seq | 11(45149 cells) |
| TCGA-LUAD | Illumina HiSeq | RNA-seq | 513 |

**Table 2. Markers used to identify epithelial cell subsets in the analysis of scRNA-seq data.**

| Cell types | Cell Markers |
|---|---|
| AT-2 | *SFTPB, SFTPC, SFTPD, ETV5, SFTPA1, ABCA3* |
| AT-1 | *CAV1, AGER, MYRF* |
| Club | *SCGB3A2* |
| Ciliated | *FOXJ1, CCDC78, PIFO, TPPP3* |
| basal | *KRT5, KRT14, TP63, KRT17, KRT6A* |
| goblet | *SPDEF, MUC5B, MUC5AC* |
| neuroendocrine | *CALCA, CHGA, ASCL1* |

threshold to screen differential genes, and then the top 5000 highly variable genes were screened from differential genes according to median absolute deviation (MAD) for consensus clustering. The R package ConsensusClusterPlus [18] was used to identify *TMSB10* associated molecular subtypes of lung adenocarcinoma. R packages sigclust was used to calculate clustering significance. A chi-square test or Fisher's exact test was used to calculate the significance of LUAD staging between molecular subtypes. Survival analysis was performed using the R package Survival and survminer. In addition, in order to characterize different molecular subtypes, wilcoxon rank sum test was used to compare the proportion of immune cells among cluster3, cluster4, cluster5 and cluster6.

## Statistical analysis

The purpose of statistical analysis was tantamount to analyze the distribution characteristics of pathological stages among molecular subtypes. The minimum theoretical frequency (T) and the total sample number (N) were used to judge the test method. If $T \geq 5$ and $N \geq 40$, pearson chi-square test were used. If $T < 5$ and $T \geq 1$ and $N > 40$, using continuity correction chi-square test, if $T < 1$ and $N < 40$, the Fisher test is used.

## Molecular characterization of *TMSB10* related molecular subtypes

We used R package WGCNA [19] to analyze the weighted gene co-expression network of LUAD expression matrix after variance-stabilizing transformation(VST). We set mergeCutHeigh = 0.25 and minModuleSize = 30. The threshold value of hub Gene was Gene significance$> 0.2$, Module Membership$>0.8$. The screening threshold of hub gene KEGG pathway analysis was p value$<0.01$.

## Prediction of *TMSB10* related molecular subtypes by random forest

We establish a random forest model to predict the molecular subtypes associated with *TMSB10*. We randomly selected 20% of the samples as the external test set, and set mtry = 41 and ntree = 500.

## Results

### *TMSB10* is regulated by super-enhancers in the epithelial-to-mesenchymal transition in lung adenocarcinoma

We assessed the distribution of super-enhancers and closest active SE-genes before and after TGF-β induction in A549 cell line. We assessed the distribution of A549 super-enhancers and their related genes before and after TGF-β induction. 1064 super-enhancers were identified in

A549 cells prior to TGFβ induction. After TGF-β induction, 530 super-enhancers were identified in the A549 cell line. There were 352 overlaps of active SE-genes closest to the super-enhancer before and after induction, while A549 had 139 SE-genes specific after induction (Fig 2a).

After GO analysis and KEGG pathway analysis of 139 specific SE-genes, we found that SE-genes were significantly enriched in HIF-1 signaling pathway, and involved biological processes such as regulation of epithelial cell migration that may affect cancer cell metastasis(Fig 2b and 2c).

We used lung adenocarcinoma scRNA-seq data (GSE131907) to exclude tumor heterogeneity. A total of six epithelial cell subsets were identified and AT-2 was utilized to investigate the difference between adenocarcinoma tumor tissue and normal lung(Fig 2d). Since the metastasis and invasion of lung adenocarcinoma cells are linked to EMT, we used the stages of lung adenocarcinoma to evaluate the progression of EMT. We used copykat to identify cancer cells in AT-2(Fig 2e), and 1,700 differential genes(p value<0.05) were obtained by analyzing the difference between stage III cancer cells and stage I cancer cells(Fig 2f).

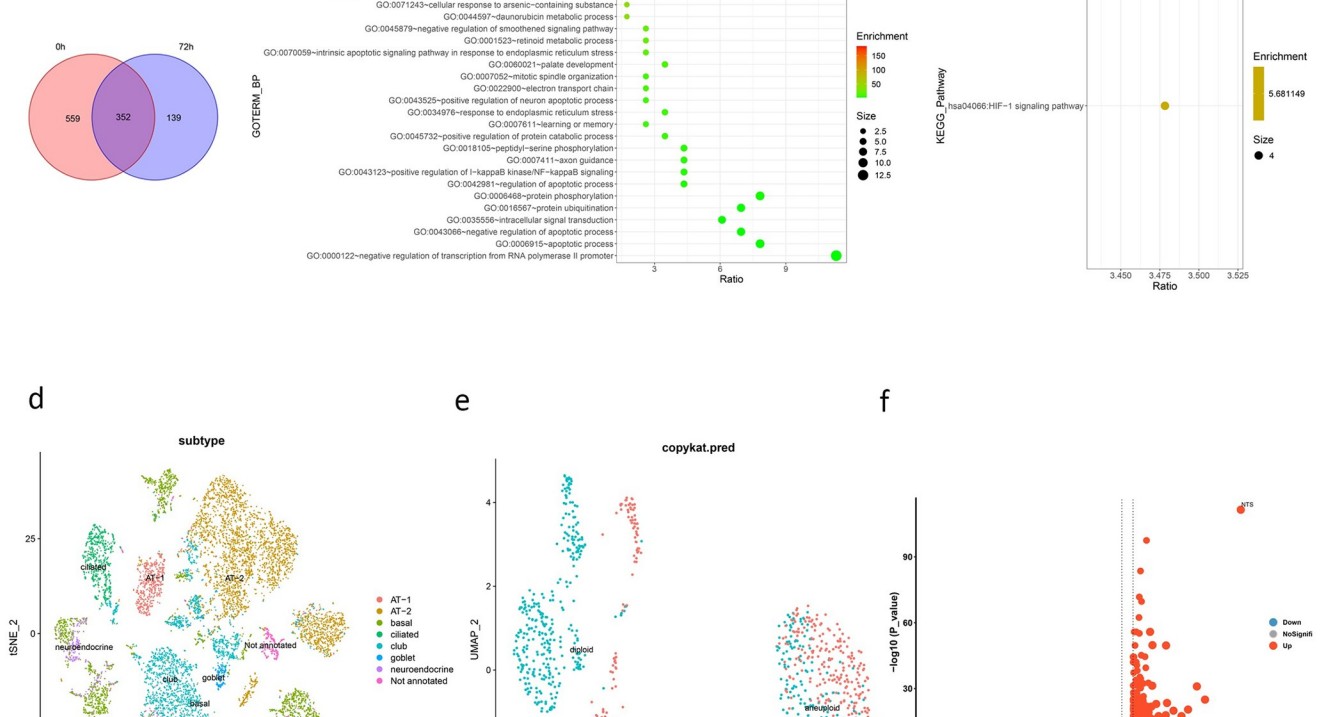

**Fig 2. Study on super-enhancer related genes. (a)** The difference of super-enhancer related genes before and after TGF-β induction, with 352 overlap genes and 139 specific genes after induction. **(b)** The GO analysis of 139 specific genes after induction. **(c)** The KEGG pathway of 139 specific genes after induction. **(d)** The cell annotation results of the scRNA-seq data of lung adenocarcinoma and normal lung. Epithelial cells were separately extracted for annotation. **(e)** The result of using copykat to identify cancer cells with AT-2 in lung adenocarcinoma samples. **(f)** The differential expression of cancer cells in stage III lung adenocarcinoma and stage I lung adenocarcinoma.

Based on the above two groups of different genes and SE-genes intersection, we get five genes: *TMSB10*, *P4HB*, *GAPDH*, *CTSB*, *NQO1*(Fig 3b). Unfortunately, the expression of *P4HB*, *CTSB* and *NQO1* is not linked with prognosis, so we will not carry out in-depth discussion. High expression of both *TMSB10* and *GAPDH* were significantly associated with poor prognosis, but we found that padj of *GAPDH* was larger and log2FC was smaller than that of *TMSB10* when we analyzed the differences among cancer cells at different stages (log2FC = 2.28 for TMSB10 and log2FC = 1.29 for *GAPDH*). Pan-cancer analysis showed that *TMSB10* was abnormally expressed in most cancers(Fig 3d), suggesting that the mechanism of action of *TMSB10* on lung adenocarcinoma may be also included in other cancers. So we focused on *TMSB10* for analysis.

## The biological significance of *TMSB10* expression level

With P value< 0.01, log2FC > | 1 | as the threshold, we identified 690 different genes expressed in *TMSB10* high and low expression group(Fig 4a). The results of GO analysis and KEGG

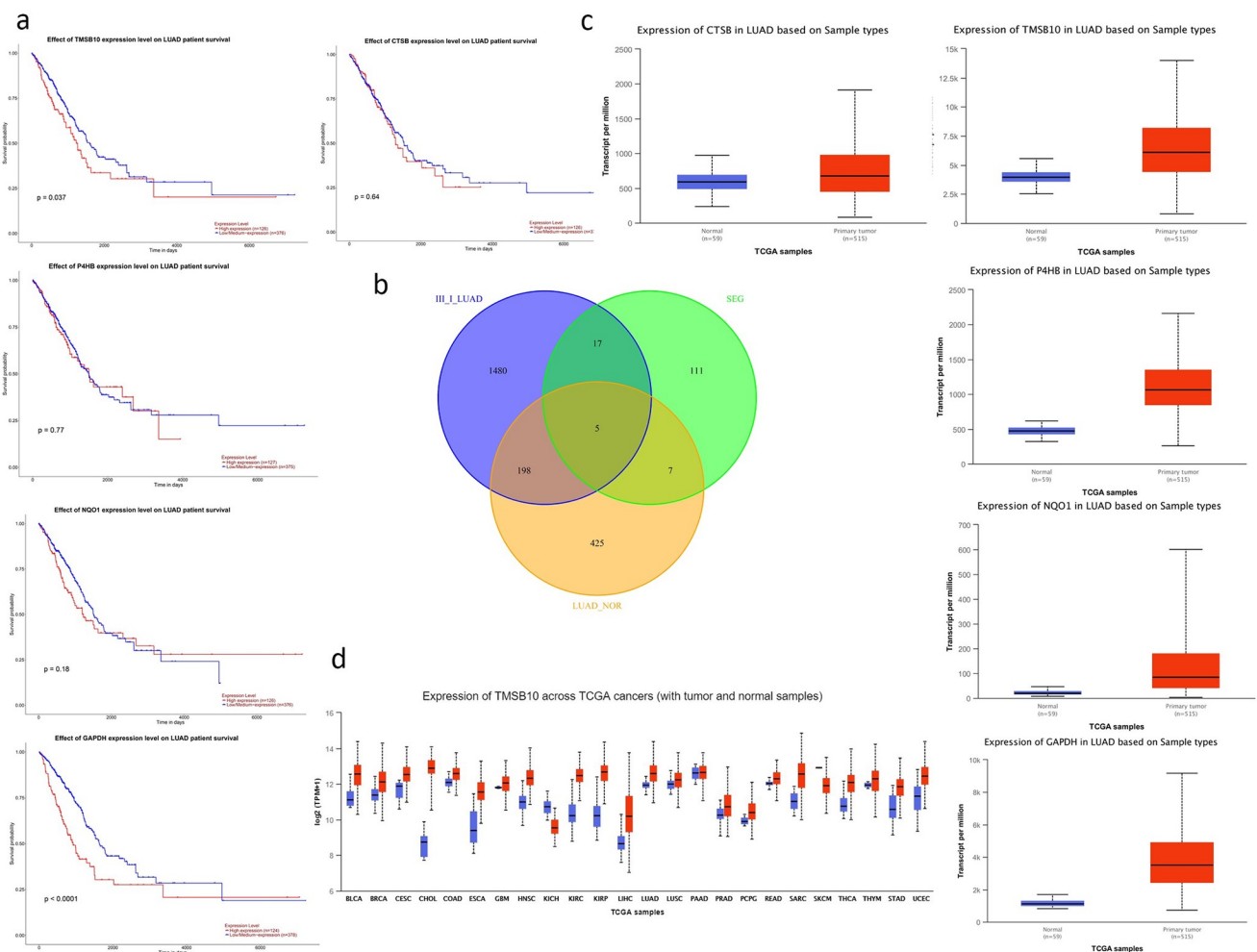

**Fig 3. TCGA data analysis results. (a)** The survival analysis result of the final screening 5 genes in the TCGA database. High expression of *TMSB10* and *GAPDH* was significantly associated with poor prognosis. **(b)** The intersection of TGF-β-induced specific SE-genes, stage III and stage I lung adenocarcinoma cell differential genes and AT-2 differential genes between lung adenocarcinoma and normal lung. We found five genes:*TMSB10*, *P4HB*, *GAPDH*, *CTSB*, *NQO1*. **(c)** The expression of 5 genes in lung adenocarcinoma samples in the TCGA database. Except for *CTSB*, the other four genes were significantly up-regulated. **(d)** The expression of *TMSB10* in a variety of cancers. *TMSB10* is significantly up-regulated in a variety of cancers.

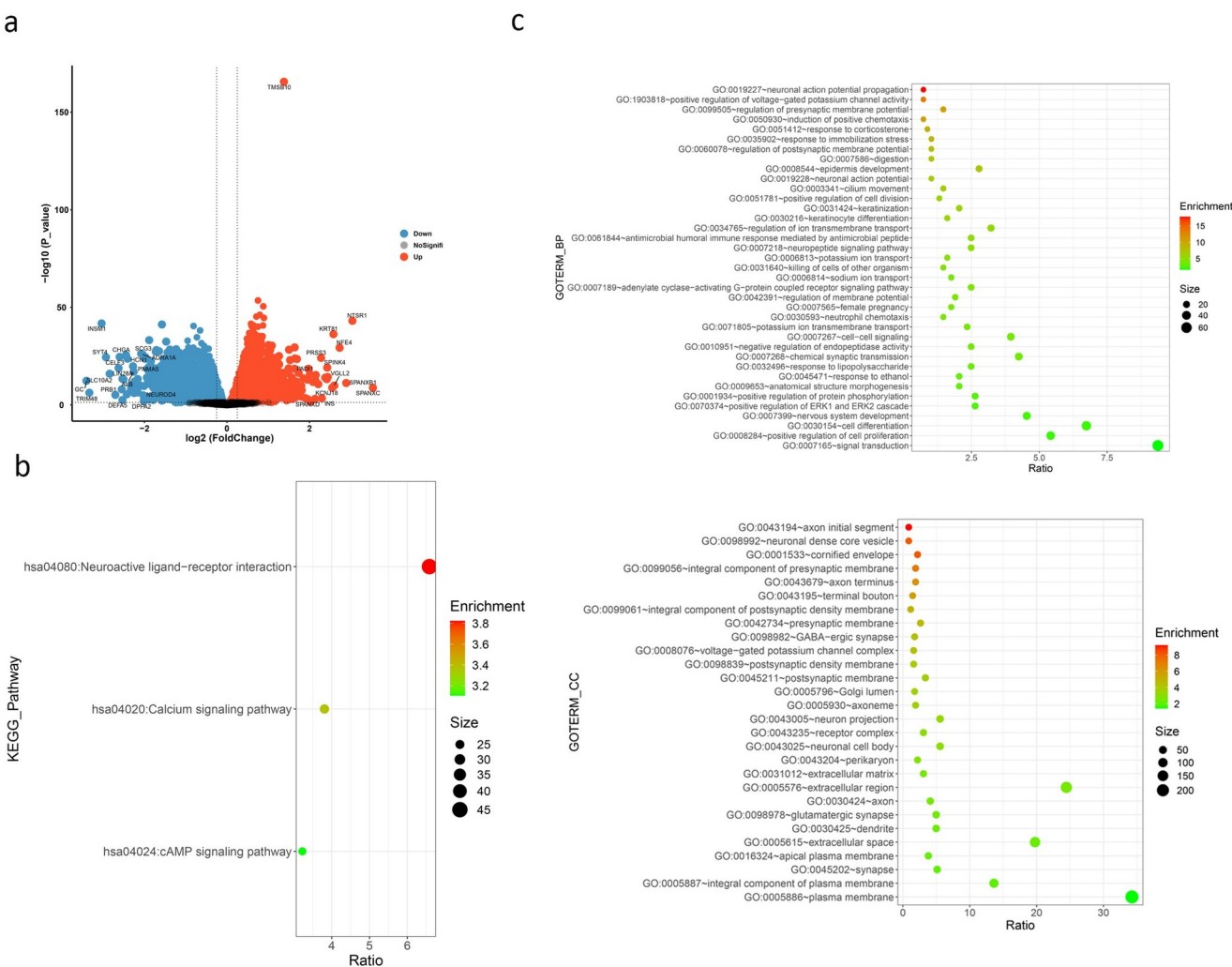

**Fig 4. Differential gene analysis between high and low *TMSB10* expression groups. (a)** The volcanic map of the difference between high and low *TMSB10* expression groups. **(b)** The result of KEGG pathway of different genes in high and low groups. **(c)** The result of GO analysis of different genes in high and low groups. We showed that with a P value of less than 0.01, differential genes were significantly enriched in the Neuroactive ligand-receptor interaction pathway, cMAP signaling pathway and other signaling pathways associated with lung cancer.

pathway showed that differential genes were enriched in Neuroactive ligand-receptor interaction pathway, Calcium signaling pathway, cAMP signaling pathway.

## The expression level of *TMSB10* revealed different characteristics of immune infiltrates

There were seven immune cell subtypes with significant differences in the proportion of *TMSB10* high expression group and low expression group: CD8 T cells, T cells CD4 memory resting, T cells CD4 memory activated, NK cells activated, marcrophages M1, mast cell resting, dendritic cells activated. The proportion of T cells CD4 memory resting and mast cell resting in the low expression group of *TMSB10* was higher than that in the high expression group, while the other five kinds of immune cells were on the contrary(Fig 5b). In addition, the expression of *TMSB10* was negatively correlated with T cells CD4 memory resting with a correlation coefficient = -0.31(Fig 5c).

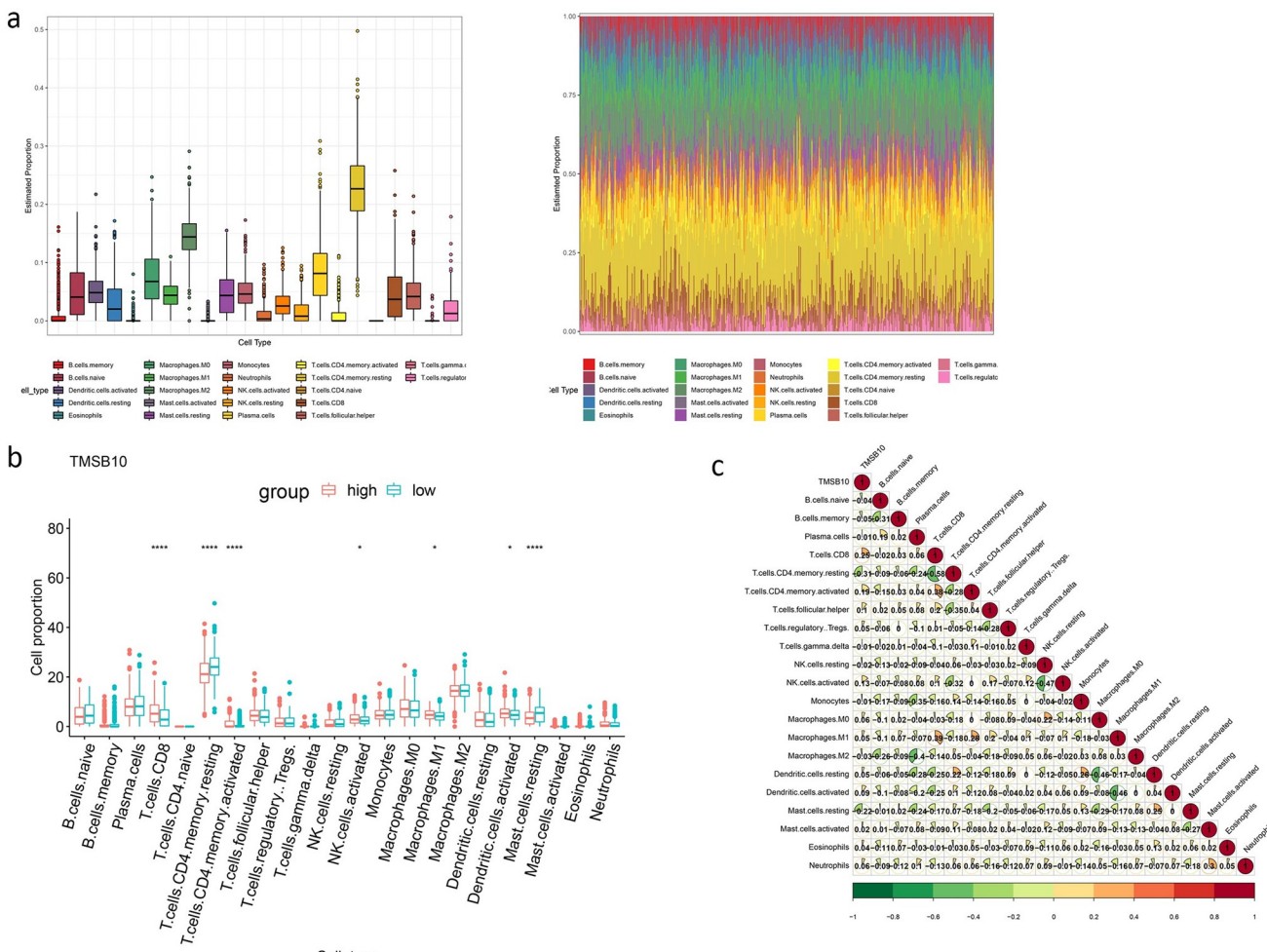

**Fig 5. Prediction of immune infiltration of lung adenocarcinoma.** P-value <0.05 was invoked as the threshold to screen samples. **(a)** The box plot showing the proportion of immune cells in different levels and the proportion of immune cells in all samples. **(b)** Difference in the proportion of immune cells between the high and low *TMSB10* expression groups. CD8, CD4 memory resting, CD4 memory activated, NK cell activated, Marcophages M1, Dendritic cells activated. Mast cells resting were significantly different between the two groups. **(c)** The correlation between *TMSB10* and different immune cells and the correlation between different immune cells.

## Identification and characterization of *TMSB10* related molecular subtypes

Using P values <0.05 as the threshold, 11,410 differential genes were screened from 19,662 genes in the high expression group and the low expression group of *TMSB10*. From the differential genes, the top 5000 variable genes were selected according to median absolute deviation (MAD) for consensus clustering. We identified eight molecular subtypes associated with *TMSB10*(Fig 6a). By comparing the survival analysis results of eight molecular subtypes, we found that cluster3 and cluster6 had a worse prognosis than other molecular subtypes(Fig 6c). We compared cluster3 and cluster6 with the extra seven subtypes respectively and found that the prognosis was not significant when only cluster3 and cluster6 were compared (P value = 0.29). Both cluster3 and cluster6 were significantly related to poor prognosis compared with other molecular subtypes (Fig 6e). Chi-square test and Fisher's exact test showed that cluster6 had different proportions of tumor stages compared with other molecular subtypes

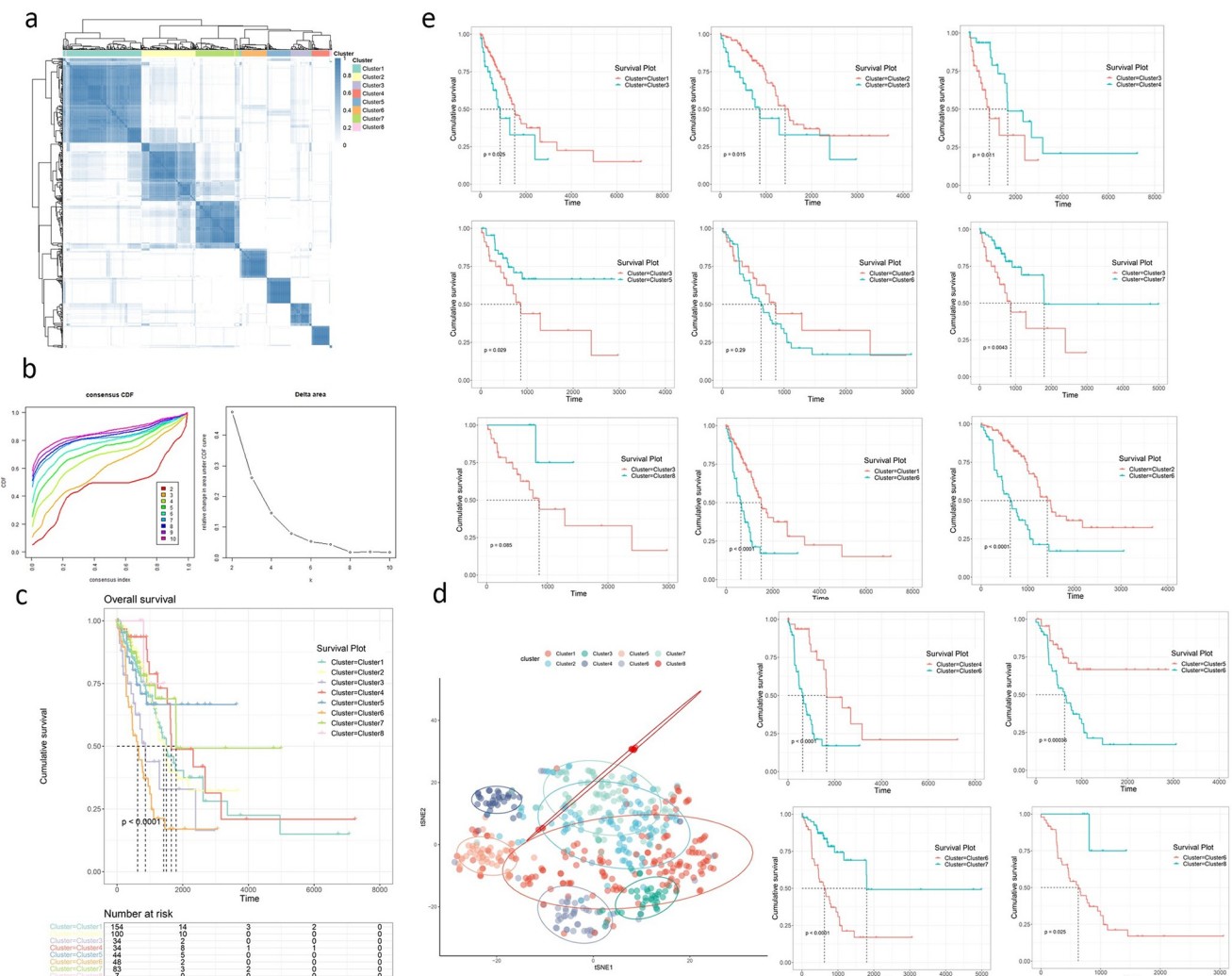

**Fig 6. Identification and characterization of molecular subtypes. (a)** we show the consensus clustering results when K = 8. **(b)** Screening method of K. **(c)** The survival analysis result of eight clusters(molecular subtypes, clusters3 and clusters6 have a poor prognosis. **(d)** The tSNE visualization results of eight clusters showed that cluster3, cluster4, cluster5, cluster6 were significantly different from other clusters, and these four clusters will be analyzed emphatically. **(e)** Prognosis of cluster3 and cluster6 compared with other subtypes.

(Tables 3 and 4). We calculated the molecular subtypes significance of eight clusters to ensure that the clustering of eight molecular subtypes was significant (S1 Table).

In addition, tSNE visualization results showed that the four subtypes of cluster3, cluster4, cluster5 and cluster6 are more independent and have better aggregation than the other molecular subtypes(Fig 6d). Therefore, we will focus on the four molecular subtypes cluster3, cluster4, cluster5 and cluster6 for subsequent analysis.

## The four *TMSB10* related molecular subtypes showed different characteristics of immune cell infiltration

We compared the proportion of immune cells among the four molecular subtypes and showed that cluster3 had a significantly increased proportion of T cell CD4 memory

**Table 3. Chi-square test of pathological stages among molecular subtypes.**

| Group1 | Group2 | p-value | Test type |
|---|---|---|---|
| Cluster1 | Cluster2 | 0.532339586 | chisq.test_correct |
| Cluster1 | Cluster3 | 0.809415309 | chisq.test_correct |
| Cluster1 | Cluster4 | 0.770325602 | chisq.test_correct |
| Cluster1 | Cluster5 | 0.721585786 | chisq.test_correct |
| Cluster1 | Cluster6 | 0.000113332 | chisq.test_correct |
| Cluster1 | Cluster7 | 0.026444948 | chisq.test_correct |
| Cluster1 | Cluster8 | 0.540429575 | fisher.test |
| Cluster2 | Cluster3 | 0.604338851 | chisq.test_correct |
| Cluster2 | Cluster4 | 0.614960072 | chisq.test_correct |
| Cluster2 | Cluster5 | 0.631191309 | chisq.test_correct |
| Cluster2 | Cluster6 | 0.00030942 | chisq.test_correct |
| Cluster2 | Cluster7 | 0.037547157 | chisq.test_correct |
| Cluster2 | Cluster8 | 0.573940463 | fisher.test |
| Cluster3 | Cluster4 | 0.543362771 | chisq.test_correct |
| Cluster3 | Cluster5 | 0.794271507 | chisq.test_correct |
| Cluster3 | Cluster6 | 0.048467852 | chisq.test_correct |
| Cluster3 | Cluster7 | 0.021586324 | chisq.test_correct |
| Cluster3 | Cluster8 | 0.406334585 | fisher.test |
| Cluster4 | Cluster5 | 0.877056475 | chisq.test_correct |
| Cluster4 | Cluster6 | 0.003993865 | chisq.test_correct |
| Cluster4 | Cluster7 | 0.564001455 | chisq.test_correct |
| Cluster4 | Cluster8 | 0.876509637 | fisher.test |
| Cluster5 | Cluster6 | 0.025503025 | chisq.test_correct |
| Cluster5 | Cluster7 | 0.107090595 | chisq.test_correct |
| Cluster5 | Cluster8 | 0.514754857 | fisher.test |
| Cluster6 | Cluster7 | 3.56E-07 | chisq.test_correct |
| Cluster6 | Cluster8 | 0.006745914 | fisher.test |
| Cluster7 | Cluster8 | 1 | fisher.test |

**Table 4. The frequency of molecular subtype six(cluster6) and the other seven subtypes.**

| | Stage I | Stage II | Stage III | Stage IV |
|---|---|---|---|---|
| Cluster1 | 87 | 36 | 28 | 5 |
| Cluster6 | 10 | 16 | 16 | 6 |
| Cluster2 | 54 | 30 | 13 | 4 |
| Cluster6 | 10 | 16 | 16 | 6 |
| Cluster3 | 17 | 10 | 8 | 1 |
| Cluster6 | 10 | 16 | 16 | 6 |
| Cluster4 | 20 | 6 | 5 | 2 |
| Cluster6 | 10 | 16 | 16 | 6 |
| Cluster5 | 22 | 10 | 8 | 3 |
| Cluster6 | 10 | 16 | 16 | 6 |
| Cluster6 | 10 | 16 | 16 | 6 |
| Cluster7 | 58 | 12 | 6 | 5 |
| Cluster6 | 10 | 16 | 16 | 6 |
| Cluster8 | 6 | 1 | 0 | 0 |

activated and macrophages M1 compared to the other three molecular subtypes. The proportion of neutrophils in cluster4 was significantly lower than in the other three molecular subtypes, and the proportion of monocytes in cluster5 is significantly lower than in other molecular subtypes. Macrophages M0 of cluster6 is significantly higher than cluster3 and cluster4 but not significantly higher than cluster5 (P = 0.052), although it is also higher (Fig 7).

In addition, as molecular subtypes significantly associated with poor prognosis, the proportions of dendritic cells resting, macrophages M1 and regulatory Treg in cluster3 and cluster6 are significantly different. Among them, cluster6 has a higher proportion of regulatory Treg than cluster3, while cluster3 has a higher proportion of macrophages M1 and dendritic cells resting than cluster6(Fig 7).

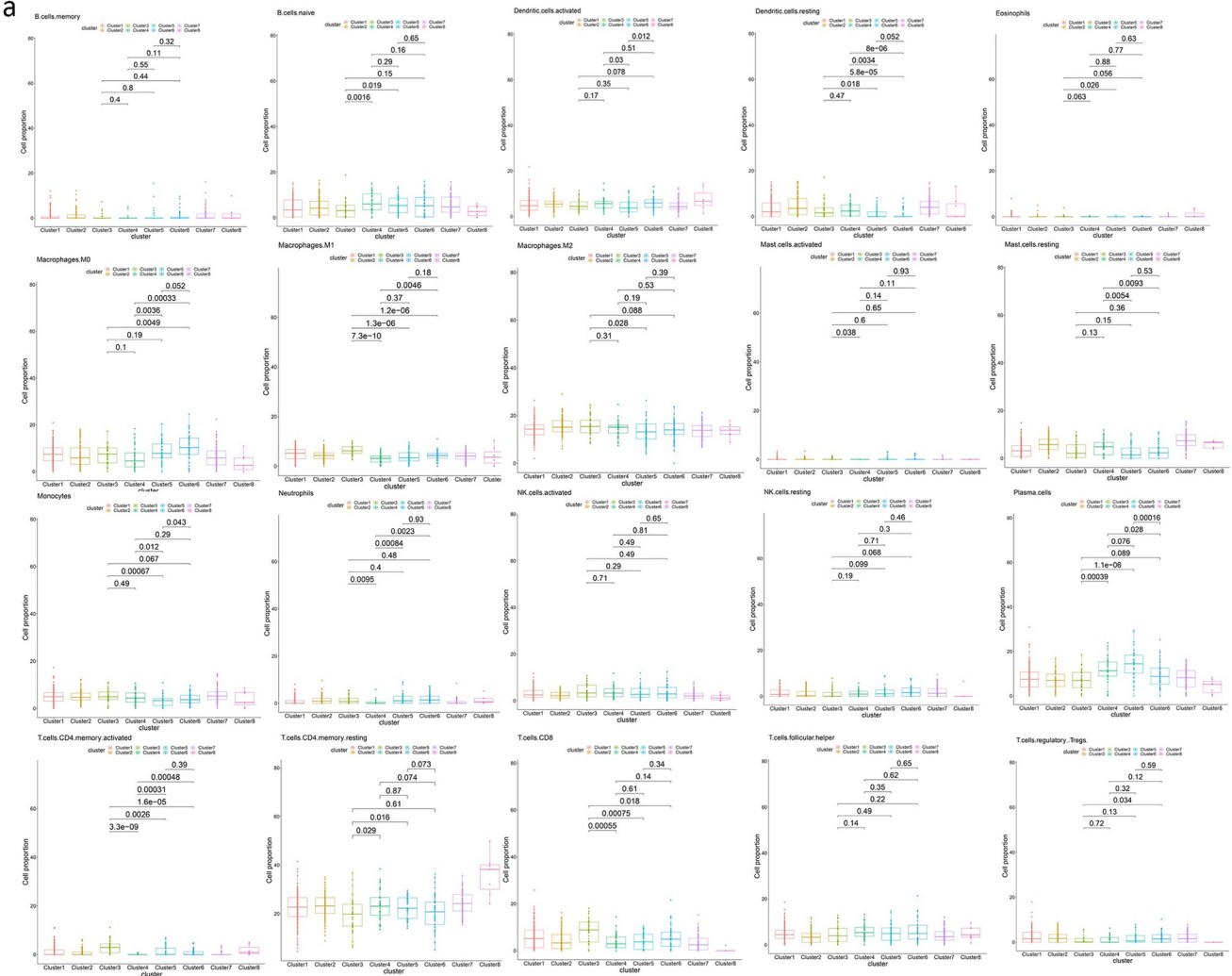

**Fig 7. Molecular subtype immune cell proportion display. (a)** We compared the proportion of immune cells in cluster3, cluster4, cluster5 and cluster6. The proportion of T cell CD4 memory activated and macrophages M1 in cluster3 was significantly higher than that in the other three clusters. The proportion of neutrophils in cluster4 was significantly lower than in the other three clusters. The proportion of monocytes in cluster5 is also significantly lower than in the other groups. Macrophages M0 of cluster6 is significantly higher than cluster3 and cluster4 but not significantly higher than cluster5 (P = 0.052), although it is also higher.

## Weighted gene co-expression network analysis revealed different molecular characteristics of the four subtypes associated with *TMSB10*

Weighted gene co-expression network analysis divided genes into 29 modules(Fig 8b). We focused on gene modules that significantly associate with four molecular subtypes. Magenta module is associated with cluster3, blue module is associated with cluster4, skyblue module is associated with cluster5, and light cyan module is associated with cluster6(Fig 8c). Hub genes screened for each module were analysed by KEGG pathway analysis. The results showed that the enrichment pathways of gene modules associated with the four molecular subtypes were specific (Fig 8f). Cluster3-related gene modules are significantly enriched in the PI3K–Akt signaling pathway, Cluster4-associated gene modules are significantly enriched in the p53

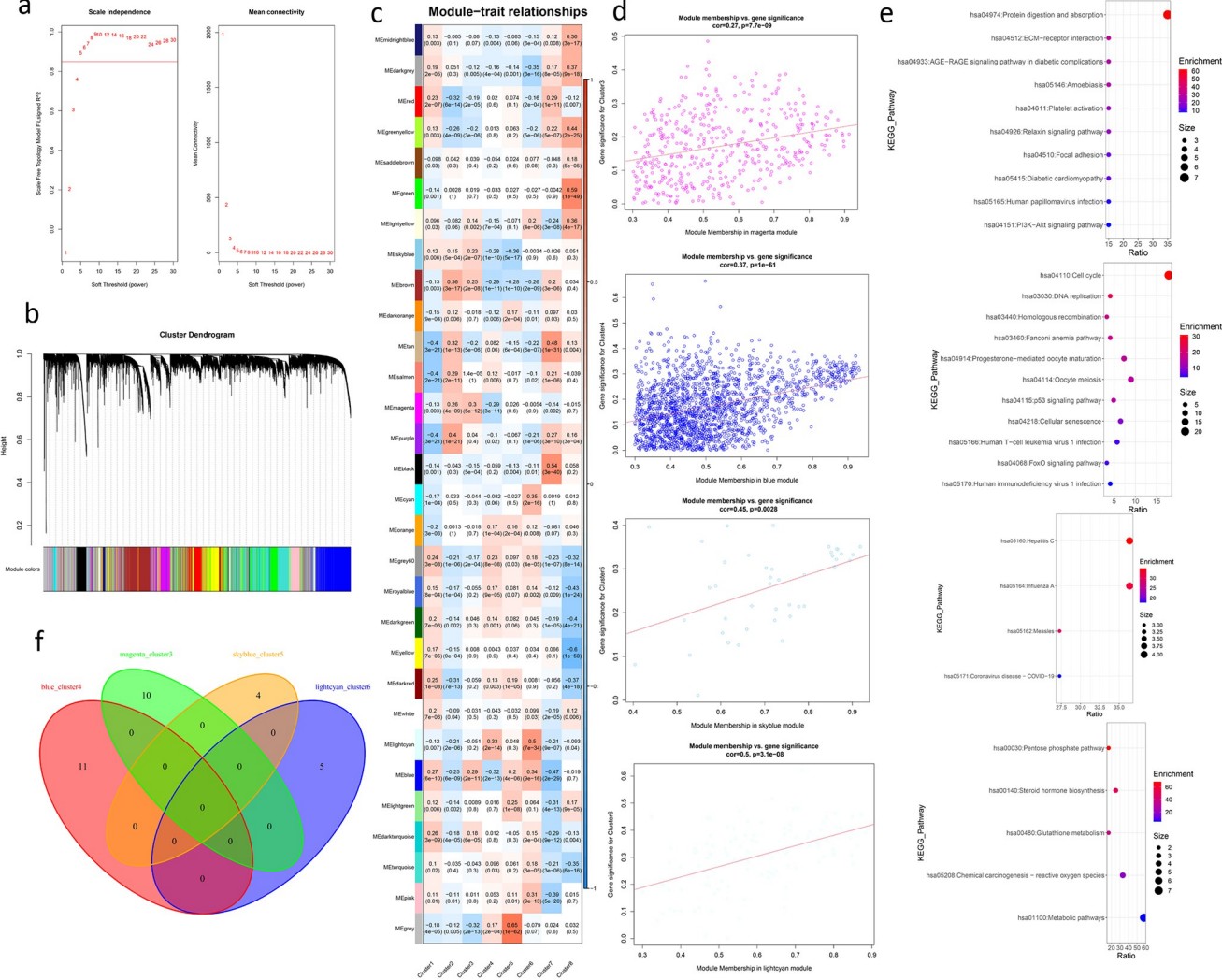

**Fig 8. Results of weighted gene co-expression network analysis. (a)** The basis for power selection. **(b)** The result of module clustering. We divided genes into 29 modules. **(c)** The correlation analysis between molecular subtypes and gene modules. **(d)** We focus on the correlation analysis between claster3 and magenta module, cluster4 and blue module, cluster5 and skyblue module, and cluster6 and lightcyan module. **(e)** Hub genes were used for GO and KEGG pathway analysis, and it was observed that the modules associated with four molecular subtypes involved in different pathways. **(f)** The intersection of KEGG pathway results of hub genes in modules associated with four molecular subtypes. The four module hub genes involve pathways that do not intersect.

signaling pathway and FoxO signaling pathway, Cluster6-related gene modules are significantly enriched in Pentose phosphate pathway and chemical carcinogenesis-reactive oxygen species, which are associated with lung cancer(Fig 8d and 8e).

## Prediction of molecular subtypes by random forest model based on hub genes

Based on the assessment of the false positive rate, we use mtry = 41 to build the random forest model(Fig 9a). And rank the importance of the genes(Fig 9b). We randomly selected 20% samples at the test set, and the results show that the prediction accuracy of external test set of random forest model is 0.87(Table 5).

## Discussion

Our research suggest that *TMSB10* may not regulate by super-enhancers in A549 cell lines that are not induced by TGF-β, and that *TMSB10* acts as a super-enhancer related activate gene only after induction of TGF-β. This suggests that *TMSB10* may be regulated by super-enhancers in lung adenocarcinoma EMT and exhibit higher expression levels based on pathological status. This phenomenon had not meant looked at in previous studies.

A549 after TGF-β induction for 72h had 139 specific SE-genes, which were significantly enriched in HIF-1 signaling pathway. HIF-1 signaling pathway has long been known and concerned because of its critical role in regulating tumor angiogenesis and promoting tumor growth, invasion and metastasis. High HIF-1αexpression predicted higher TNM staging, distant metastasis, and vascular invasion [20].

Pan-cancer analysis showed that *TMSB10* was differentially expressed in most cancers, most of which were highly expressed. This suggests that the mechanism by which *TMSB10* is highly expressed in lung adenocarcinoma may also apply to other cancers (Fig 3d).

KEGG pathway analysis revealed the biological significance of the expression level of *TMSB10*(Fig 4b). Neuroactive ligand-receptor interaction pathway is strongly associated with smoking-induced lung cancer [21].

Each of the four *TMSB10*-related molecular subtypes we identified had its own characteristics for immune cell infiltration. Cluster3 has a higher proportion of CD4 memory activated and macrophages M1, cluster4 has a low proportion of neutrophils, and cluster5 has a low proportion of monocytes. The proportion of cluster6 macrophages M0 is greater than other subtypes.

In the above results, macrophages were associated with cluster3(M1) and cluster6(M0). Macrophages M2 and small populations of M1 cells, also known as tumor-associated macrophages (TAMs). TAMs in tumor microenvironment promote tumor proliferation and the formation of intratumor blood vessels, promote tumor cell invasion and metastasis, and are associated with poor prognosis and drug resistance of cancer [22]. There was no difference in the proportion of macrophage M2 among four subtypes except that cluster3 was significantly higher than cluster5. The proportion of macrophage M1 promoting an inflammatory response was higher in cluster3. However, macrophage M1 is a tumor suppressor, suggesting that the poor prognosis of cluster3 may be due to other causes. This requires further research.

Hub genes of the gene modules associated with the molecular subtype can usually reflect the molecular characteristics of the molecular subtype. Results of KEGG pathway analysis of hub genes of the four gene modules did not overlap. This suggests that the pathways associated with each of the four molecular subtypes are specific.

Hub genes of Cluster3-related gene module magenta and Cluster4-related gene module blue are significantly enriched in the PI3K/Akt signaling pathway and p53 signaling

a

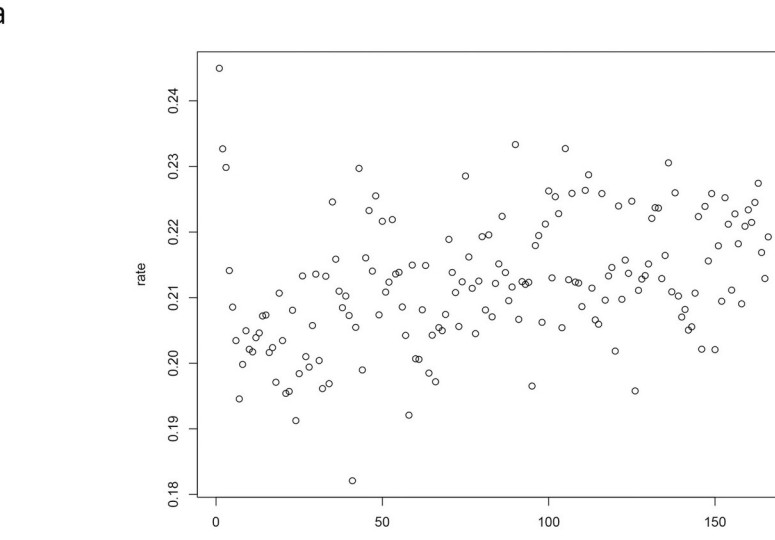

importance

b

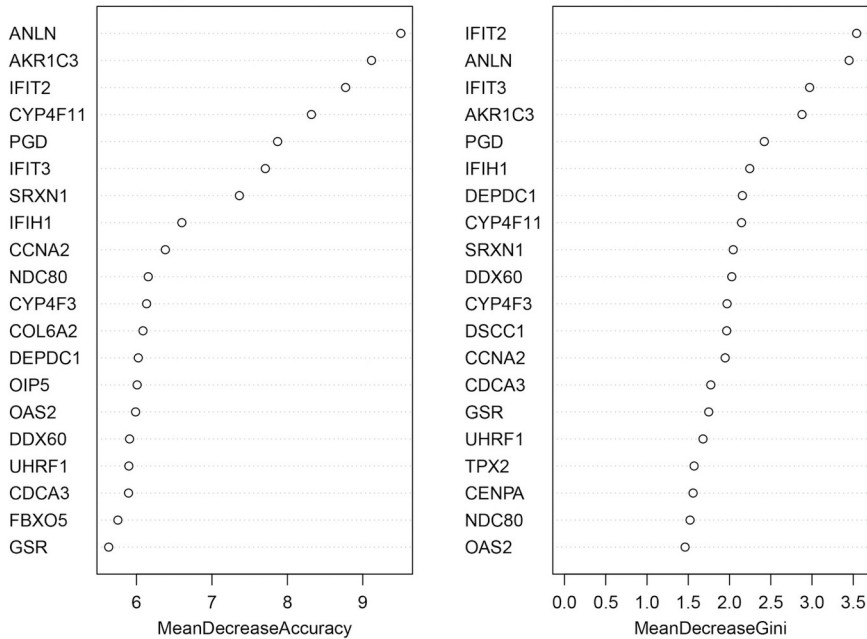

**Fig 9. Parameter selection and importance ranking of stochastic forest prediction model.** (**a**) Mtry parameter adjustment. The value of mtry was determined by calculating the mean value of the model's misjudgment rate. The results showed that when mtry = 41, the model's misjudgment rate was the lowest, which was 0.182. (**b**) The top 20 in the importance ranking of hub genes involved in the establishment of random forest model.

pathway respectively, which are well-studied pathways in non-small cell lung cancer and can promote proliferation and inhibit apoptosis. In addition, PI3K/AKT pathway activation may be a potential mechanism of chemoradiotherapy resistance in small cell lung cancer [23].

**Table 5. Confusion matrix for prediction of external test sets by random forest model.**

|          | Cluster3 | Cluster4 | Cluster5 | Cluster6 |
|----------|----------|----------|----------|----------|
| Cluster3 | 3        | 0        | 0        | 1        |
| Cluster4 | 0        | 5        | 0        | 0        |
| Cluster5 | 0        | 1        | 8        | 0        |
| Cluster6 | 0        | 0        | 2        | 10       |

In the four molecular subtypes associated with *TMSB10*, we noted that cluster6 had the worst prognosis. Hub genes of Cluster6-associated gene module lightcyan are significantly enriched in the pentose phosphate pathway and chemically carcinogenic reactive oxygen species, both of which are associated with lung cancer(Fig 8d and 8e). The abnormal accumulation of *ROS* in lung adenocarcinoma is due to the imbalance of *REDOX* state caused by down-regulation of the pentose phosphate pathway (PPP) and inactivation of fatty acid oxidation (FAO) pathway [24], suggesting that the two pathways may interact.

Reactive oxygen species play different roles before and after cancer. In the precancerous state, reactive oxygen species contribute to the development of cancer through oxidative stress and base-pair replacement mutations in proto-oncogenes and tumor suppressor genes. They aid in cancer invasion and metastasis by activating the NF-kB and MAPK pathways in later stages of tumor progression [25]. In another study, pentose phosphate pathway flux was also shown to be a basic metabolic pathway supporting cancer cell growth and invasion [26]. A drug study has shown that Ginseng-derived nanoparticles inhibit lung cancer cell EMT by inhibiting the activity of the pentose phosphate pathway [27]. This suggests that molecular subtype characteristics of Cluster6 may be closely related to EMT, but what exactly causes the poor prognosis needs further research.

## Conclusions

In conclusion, our study found that *TMSB10* may be regulated by super-enhancers in EMT of lung adenocarcinoma cells. Inhibition of super-enhancers regulating *TMSB10* may be a new way to treat lung adenocarcinoma. We revealed the different molecular, immune cell infiltration, and pathological stages characteristics of the four molecular subtypes associated with *TMSB10*. In order to identify distinct molecular subtypes in clinical diagnosis, we established a random forest prediction model based on hub genes, and the prediction accuracy was 0.87. This may provide potential help for personalized medicine and diagnosis of lung adenocarcinoma.

## Supporting information

**S1 Table. Clustering significance of molecular subtypes.**
(XLSX)

## Author Contributions

**Formal analysis:** Yifei Liu.

**Methodology:** Yifei Liu.

**Visualization:** Yifei Liu.

**Writing – original draft:** Yifei Liu.

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
