## [Decision Letter · Decision Letter 0]

5 Apr 2023

PONE-D-23-03016Identification and comprehensive analysis of super-enhancer related genes involved in epithelial-to-mesenchymal transition in lung adenocarcinomaPLOS ONE

Dear Dr. Yifei,

Thank you for submitting your manuscript to PLOS ONE. After careful consideration, we feel that it has merit but does not fully meet PLOS ONE’s publication criteria as it currently stands. Therefore, we invite you to submit a revised version of the manuscript that addresses the points raised during the review process.

We look forward to receiving your revised manuscript.

Kind regards,

Gurudeeban Selvaraj

Academic Editor

PLOS ONE

“The work was supported by Natural Science Foundation of Fujian Province (2019J01474).

URL:http://xmgl.kjt.fujian.gov.cn/showLoginPage.do?type=fujian&loginflag=false. Funders provided financial support.”

Reviewers' comments:

Reviewer's Responses to Questions

**Comments to the Author**

1. Is the manuscript technically sound, and do the data support the conclusions?

Reviewer #1: Yes

2. Has the statistical analysis been performed appropriately and rigorously? 

Reviewer #1: N/A

3. Have the authors made all data underlying the findings in their manuscript fully available?

Reviewer #1: Yes

4. Is the manuscript presented in an intelligible fashion and written in standard English?

Reviewer #1: Yes

5. Review Comments to the Author

Reviewer #1: I read this paper with a lot of interest. The author’s main goal is to understand the mechanism of lung adenocarcinoma which causes high mortality rate .The study gives the hypothesis that the up-regulation of TMSB10 in lung adenocarcinoma and its connection with poor prognosis may be caused by the regulation of super-enhancers during tumour cell metastasis by examining the single-cell sequencing data and the TCGA database of lung cancer. The work is properly organized and presented, the procedures are given clearly, and the findings are thoroughly addressed. The title of the paper “Identification and comprehensive analysis of super-enhancer related genes involved in epithelial-to-mesenchymal transition in lung adenocarcinoma” is well supported by the data presented. This article is an important addition to the literature.

Before accepting the paper, I have few serious concerns to be addressed:

1. The materials and methods section is informative and well organized however, I would suggest to provide a schematic diagram or flowchart of the workflow.

2. A separate table for the datasets could be included in the manuscript with details about the platform, expression type, and sample count (high/low).

3. If there are any potential batch effects brought on by data variations (SEdb, GEO), authors could address these.

4. Multiple testing correction is crucial in identifying differential gene expression, although the authors merely used the P value as a criterion in this investigation.

5. If the author used P value to determine the top 5000 variables for clustering analysis, they should state how many variables in total there were.

6. PLOS authors have the option to publish the peer review history of their article (what does this mean?). If published, this will include your full peer review and any attached files.

Reviewer #1: No

---

## [Author Response · Author response to Decision Letter 0]

8 Apr 2023

Dear editor and reviewers:

Thank you for your letter and the reviewers’ comments on our manuscript entitled “Identification and comprehensive analysis of super-enhancer related genes involved in epithelial-to-mesenchymal transition in lung adenocarcinoma” (ID: PONE-D-23-03016). Those comments are very helpful for revising and improving our paper, as well as the important guiding significance to other research. We have studied the comments carefully and made corrections which we hope meet with approval. The main corrections are in the manuscript and the responds to the reviewers’ comments are as follows.

Replies to the reviewers’ comments:

Reviewer #1:

1.The materials and methods section is informative and well organized however, I would suggest to provide a schematic diagram or flowchart of the workflow.

Response: Thanks for the reviewer's reminding. This is a very useful suggestion, and we have added the flowchart of the workflow to the manuscript. To make it easier for the reader to understand our research, we've shown flowchart as the first figure of the manuscript.

2.A separate table for the datasets could be included in the manuscript with details about the platform, expression type, and sample count (high/low).

Response: We have added a table to the manuscript to show the information of data set. SEdb 2.0 uses raw data from multiple public databases, therefore, we listed the number of the data set in the public database in this manuscript, not the number in the SEdb 2.0.

3.If there are any potential batch effects brought on by data variations (SEdb, GEO), authors could address these.

Response: Batch effects are also a concern for us. Although the analysis results of super-enhancers in SEdb 2.0 may come from different data sets, the identification of super-enhancers and their regulatory genes is based on the identification of typical enhancers by the peak of H3K27ac, and then the integration of typical enhancers to identify the super-enhancers and the genes in their regulatory range. In this process, we usually focus on the changes in the distribution position of super-enhancers on the genome under different experimental conditions, and did not involve the comparison of gene expression or other values between samples, nor did we involve statistical tests between groups. Therefore, batch effect and sample size have little influence on the identification of super-enhancers. We used the GEO dataset(GSE131907) for single cell RNA-seq analysis of lung adenocarcinoma. All samples were from the same data set, and as we mentioned in Materials and methods, we use IntegrateData() to integrate samples and remove batch effects.

4.Multiple testing correction is crucial in identifying differential gene expression, although the authors merely used the P value as a criterion in this investigation.

Response: Three of our studies involved differential gene analysis. The first was an analysis of the difference between cancer cells transformed from AT-2 in lung adenocarcinoma with stage III and lung adenocarcinoma with stage I, and the second was an analysis of the difference in AT-2 between adenocarcinoma and normal lung tissue. Through the integration analysis of the two difference analysis and the identification results of super-enhancers, we screened out TMSB10 for the focus of discussion. In the two differential expression analyses, the corrected P-value (Padj) of TMSB10 as a differential gene was 0.035 and 2.54E-37, respectively, which were both significant. Indeed, Multiple testing correction is more rigorous in filtering differential genes, but TMSB10 will eventually be screened out for study regardless of whether it is filtered with P value or corrected P-value. Although this has no impact on our research results, the suggestions put forward by the reviewers will be beneficial to our future research.

In the screening of differential genes in the high-expression and low-expression groups of TMSB10, we selected p-value as the screening criterion, because we wanted to retain as many available genes as possible for secondary screening before consensus clustering. Prior to consensus clustering, we performed a second gene filter. Finally, top 5000 highly variable genes measured by median absolute deviation (MAD) were used for analysis. We noticed that the statement about screening 5000 genes for consensus clustering was vague in the manuscript, so we revised it. 

5.If the author used P value to determine the top 5000 variables for clustering analysis, they should state how many variables in total there were. 

Response: There are a total of 19662 genes in the LUAD data set of TCGA, and 11410 differential genes are selected by P-value<0.05, while 10827 differential genes are selected by Padj<0.05. Among the 11410 differential genes screened using P-values. The top 5000 highly variable genes were selected by median absolute deviation (MAD) for consensus clustering. Thanks to the reviewer's reminding, we have also added the description of variables into the manuscript.

Once again, thank you very much for your constructive comments and suggestions which would help us in depth to improve the quality of the paper.

Kind regards.

Yifei Liu

E-mail: 2217579236@qq.com

Corresponding author : Yifei Liu

E-mail address: 2217579236@qq.com

---

## [Decision Letter · Decision Letter 1]

22 Aug 2023

Identification and comprehensive analysis of super-enhancer related genes involved in epithelial-to-mesenchymal transition in lung adenocarcinoma

PONE-D-23-03016R1

Dear Dr. Yifei,

We’re pleased to inform you that your manuscript has been judged scientifically suitable for publication and will be formally accepted for publication once it meets all outstanding technical requirements.

Kind regards,

Jinhui Liu

Academic Editor

PLOS ONE

Additional Editor Comments (optional):

I think this manuscript was well organized and it could be accepted.

Reviewers' comments:

Reviewer's Responses to Questions

**Comments to the Author**

1. If the authors have adequately addressed your comments raised in a previous round of review and you feel that this manuscript is now acceptable for publication, you may indicate that here to bypass the “Comments to the Author” section, enter your conflict of interest statement in the “Confidential to Editor” section, and submit your "Accept" recommendation.

Reviewer #2: All comments have been addressed

Reviewer #3: All comments have been addressed

2. Is the manuscript technically sound, and do the data support the conclusions?

Reviewer #2: Yes

Reviewer #3: Yes

3. Has the statistical analysis been performed appropriately and rigorously? 

Reviewer #2: Yes

Reviewer #3: Yes

4. Have the authors made all data underlying the findings in their manuscript fully available?

Reviewer #2: Yes

Reviewer #3: Yes

5. Is the manuscript presented in an intelligible fashion and written in standard English?

Reviewer #2: Yes

Reviewer #3: Yes

6. Review Comments to the Author

Reviewer #2: (No Response)

Reviewer #3: The returned revised manuscript has a improved quality and adequately responded to the questions rauised during the previous review. A super-enhancer regulated gene TMSB10 has been identified to be upregulated in lung adenocarcinoma; its role in EMT and profile of genes induced were investigated in the present study. Presentation of the data and the paper writing have been well improved in the revision. An acceptance is therefore recommended as judged by the criteria of publication set by PlosOne.

7. PLOS authors have the option to publish the peer review history of their article (what does this mean?). If published, this will include your full peer review and any attached files.

Reviewer #2: No

Reviewer #3: No

---

## [Editor Report · Acceptance letter]

25 Aug 2023

PONE-D-23-03016R1 

Identification and comprehensive analysis of super-enhancer related genes involved in epithelial-to-mesenchymal transition in lung adenocarcinoma 

Dear Dr. Liu:

I'm pleased to inform you that your manuscript has been deemed suitable for publication in PLOS ONE. Congratulations! Your manuscript is now with our production department. 

Kind regards, 

on behalf of

Dr. Jinhui Liu 

Academic Editor

PLOS ONE